# Atherosclerosis, Cardiovascular Disease, and COVID-19: A Narrative Review

**DOI:** 10.3390/biomedicines11041206

**Published:** 2023-04-18

**Authors:** Carles Vilaplana-Carnerero, Maria Giner-Soriano, Àngela Dominguez, Rosa Morros, Carles Pericas, Dolores Álamo-Junquera, Diana Toledo, Carmen Gallego, Ana Redondo, María Grau

**Affiliations:** 1Fundació Institut Universitari per a la Recerca a l’Atenció Primària de Salut Jordi Gol i Gurina (IDIAPJGol), 08007 Barcelona, Spain; 2Department of Medicine, School of Medicine and Health Sciences, University of Barcelona, 08036 Barcelona, Spain; 3School of Medicine, Universitat Autònoma de Barcelona, 08193 Bellaterra, Spain; 4Biomedical Research Consortium in Epidemiology and Public Health (CIBERESP), 28029 Madrid, Spain; 5Biomedical Research Consortium in Infectious Diseases (CIBERINFEC), 28029 Madrid, Spain; 6Department of Pharmacology, Therapeutics and Toxicology, School of Medicine, Universitat Autònoma de Barcelona, 08193 Bellaterra, Spain; 7Epidemiology Service, Public Health Agency of Barcelona (ASPB), 08023 Barcelona, Spain; 8Quality, Process and Innovation Direction, Vall d’Hebron Hospital Universitari, Vall d’Hebron Barcelona Hospital Campus, 08035 Barcelona, Spain; 9Health Services Research Group, Vall d’Hebron Institut de Recerca (VHIR), Vall d’Hebron Hospital Universitari, Vall d’Hebron Barcelona Hospital Campus, 08035 Barcelona, Spain; 10Methodology, Quality and Medical Care Assessment Department, Direcció d’Atenció Primària Metropolitana Sud, Catalan Institute of Health (ICS), 08908 Barcelona, Spain; 11Hospital Universitario Bellvitge, Catalan Institute of Health (ICS), 08907 Barcelona, Spain; 12Serra Húnter Fellow, Department of Medicine, School of Medicine and Health Sciences, University of Barcelona, 08036 Barcelona, Spain; 13August Pi i Sunyer Biomedical Research Institute (IDIBAPS), 08036 Barcelona, Spain

**Keywords:** COVID-19, atherosclerosis, cardiovascular disease, long COVID, bidirectional link, cardiovascular disease treatments

## Abstract

Atherosclerosis is a chronic inflammatory and degenerative process that mainly occurs in large- and medium-sized arteries and is morphologically characterized by asymmetric focal thickenings of the innermost layer of the artery, the intima. This process is the basis of cardiovascular diseases (CVDs), the most common cause of death worldwide. Some studies suggest a bidirectional link between atherosclerosis and the consequent CVD with COVID-19. The aims of this narrative review are (1) to provide an overview of the most recent studies that point out a bidirectional relation between COVID-19 and atherosclerosis and (2) to summarize the impact of cardiovascular drugs on COVID-19 outcomes. A growing body of evidence shows that COVID-19 prognosis in individuals with CVD is worse compared with those without. Moreover, various studies have reported the emergence of newly diagnosed patients with CVD after COVID-19. The most common treatments for CVD may influence COVID-19 outcomes. Thus, their implication in the infection process is briefly discussed in this review. A better understanding of the link among atherosclerosis, CVD, and COVID-19 could proactively identify risk factors and, as a result, develop strategies to improve the prognosis for these patients.

## 1. Introduction

From the beginning of the outbreak in late 2019, severe acute respiratory syndrome coronavirus 2 (SARS-CoV-2) infection has spread all over the world with more than 600 million cases and more than 6 million deaths [1]. The signs and symptoms of coronavirus disease (COVID-19) range from fever or fatigue, mainly in the acute nonsevere forms, to systemic complications (e.g., pulmonary, cardiovascular, endocrinological, neurological, and psychiatric symptoms) in more severe forms. These signs and symptoms could last 6 months and more, in a form called long COVID [2,3,4,5].

A growing body of evidence has shown that SARS-CoV-2 and other respiratory viruses, such as influenza, may lead to cardiovascular damage and the consequent disease onset [6,7,8,9,10,11]. Atherosclerosis is the physiopathological injury that comes before most cases of CVDs. It is a chronic inflammatory and degenerative process that mainly occurs in large- and medium-sized arteries and is morphologically characterized by asymmetric focal thickenings of the innermost layer of the artery, the intima [12]. The accumulation of fatty and/or fibrous material in the intima layer of arteries forms an atherosclerotic plaque or atheroma. As time progresses, atheroma becomes fibrous and accumulates calcium minerals. In this stage, the plaque can encroach upon the arterial lumen reducing drastically the blood flow and leading to tissue ischemia or provoking the formation of a thrombus that can also occlude the lumen causing a more acute ischemia [13]. A pro-inflammatory and thrombophilic state is an integral feature of atherosclerosis, potentially increasing vulnerability to severe COVID-19 because the underlying endothelial dysfunction might represent the ideal deregulated immunological setting in which SARS-CoV-2 triggers a “cytokine storm” [14]. In addition, atherosclerosis causes coronary artery calcifications when evolved and becomes the main cause of CVDs [15].

A confirmatory analysis has observed that influenza might contribute substantially to the burden of ischemic heart disease, the most common diseases within the group of CVDs [16]. This association has also been explored for SARS-CoV-2 infection, with several scientific works pointing out an increased incidence of myocardial injury, acute coronary syndrome, thromboembolism, and other CVDs [6,17]. This relationship could be explained by the role of these infections in the development of atherosclerosis, driven by an endothelial dysfunction caused by the virus [10]. In the particular case of SARS-CoV-2, the endothelial cells and cardiac pericytes express abundant angiotensin converting enzyme 2 (ACE2). SARS-CoV-2 uses this enzyme to facilitate the entry to target cells and to initiate infection mediated by transmembrane serine protease 2 (TMPRSS2) and cathepsin L [18]. Once inside, the virus disrupts the immune system, by a hyperinflammatory state, by the renin–angiotensin–aldosterone system, by activating vasoconstriction, inflammation and fibrotic pathways, and thrombotic balance via a prothrombotic environment caused by the activation of the coagulation cascade and the suppression of fibrinolytic mechanisms. The overall dysregulation of the endothelial injury results in a vicious cycle that causes end-organ dysfunction affecting cardiovascular, pulmonary, and immune systems [19]. As a result, a correlation among inflammatory response, COVID-19, atherosclerosis, and the clinical forms of CVDs, such as acute coronary syndromes, has been observed. Indeed, several manuscripts describe a bidirectional cause–effect relationship between COVID-19 and atherosclerosis [11,17,20,21]. Then, inflammatory and cytokine pathways of COVID-19 and atherosclerosis present similarities. Endothelial dysfunction, hyperinflammation, and coagulopathy are common in persons infected with SARS-CoV-2 and are also prevalent features of atherosclerosis [22] (Figure 1). On the one hand, the history of previous CVDs increases the risk of severe disease and death in individuals virally infected [9,23]. On the other hand, COVID-19 itself can induce myocardial injury, arrhythmia, acute coronary syndrome, venous thromboembolism, and accelerating CVD [24,25]. Thus, pharmacological agents used for atherosclerotic conditions and CVDs can modify the association in both directions [21,22,26].

The objectives of this narrative review are (1) to provide an overview of the most recent studies that point out a bidirectional relation between COVID-19 and atherosclerosis and (2) to summarize the impact of cardiovascular drugs on COVID-19 outcomes.

## 2. Atherosclerosis and CVD as COVID-19 Risk Factors

From the initial stages of the COVID-19 outbreak, studies have stated that a previous or underlying history of CVD is directly associated with a worse prognosis and severity of COVID-19 [26,27,28,29,30,31,32,33,34]. This association was still existent in a subgroup analysis by sex, region, disease type, disease outcome, and study design, suggesting stable findings [35]. In addition, individuals with cardiovascular risk factors have more risk of developing severe COVID-19 and have a higher risk of admission to the intensive care unit (ICU) and hospital death. The role of cardiovascular risk factors on mortality is accumulative, and the greater number of such factors, the higher the risk of mortality and the poorer the prognosis in individuals diagnosed with COVID-19. Thus, careful consideration is required in the mutual management of CVD and COVID-19 [33,36,37,38,39,40]. Sheppard et al. in a cohort of more than 45,000 individuals, showed the cumulative effect of hypertension in COVID-19 patients. Older patients with controlled blood pressure diagnosed for longer had a higher risk of death by COVID-19 than those with a recent diagnosis of hypertension and uncontrolled blood pressure. The authors hypothesized that older individuals could have more advanced atherosclerosis and target organ damage that predispose to worse COVID-19 outcomes [41]. Regarding the lipid profile, the results from a meta-analysis showed that a significant decrease in the levels of total cholesterol, high-density lipoprotein (HDL) cholesterol, and low-density lipoprotein (LDL) cholesterol are associated with severity and mortality in COVID-19 patients. Hence, the lipid profile may be used for assessing the severity and prognosis of COVID-19 [42]. In another study conducted in a large cohort of more than 28,000 subjects without previous CVD and a positive SARS-CoV-2 test, the high estimates of the 10-year atherosclerotic cardiovascular disease (ASCVD) risk score measured continuously were associated with a greater likelihood of severe COVID-19 [43].

Some initial hypotheses behind the high prevalence of CVD in COVID-19 patients are related with advanced age, functionally impaired immune system, or elevated levels of ACE2 [28]. In addition, an unbalanced immune response and hyperinflammatory process caused by the viral infection leads to vascular damage, such as vasculitis. The storm of cytokines and other inflammatory mediators, such as tumor necrosis factor α (TNF-α); interleukins (ILs) 1, 6, 8, and others; neutrophil activation traps (NETs); and toll-like receptor (TLR) or macrophages result in the production of reactive oxidative species (ROS) [17,19,22]. Thus, in individuals with a previous history of CVD, their inflammation would be accelerated and finally could be more affected by COVID-19 [17] (Figure 1).

Additionally, subclinical atherosclerosis can impact the course of COVID-19. Coronary artery calcification, a specific imaging marker of coronary atherosclerosis that correlates with the plaque burden, can reveal previously undiagnosed CVD in COVID-19 patients [44]. In a retrospective study of 457 individuals without a history of clinical coronary artery disease who underwent chest CT imaging during COVID-19 hospitalization, coronary artery calcification was detected in 42.9% of the patients. The presence of coronary artery calcification was associated with mechanical ventilation (*p* = 0.01), ICU admission (*p* = 0.02), in-hospital or 30-day mortality (*p* < 0.01), and hospital length of stay (*p* < 0.001) [45]. Moreover, a recent meta-analysis that included eight studies showed an increase in mortality in the presence of coronary artery calcifications (relative risk 2.24, 95% confidence interval, 1.41–3.56; *p* < 0.001) [46].

Table 1 includes the prevalence of several cardiovascular risk factors and CVDs in individuals hospitalized with COVID-19 analyzed in three different systematic reviews and meta-analyses [31,47,48]. Despite the differences found in all three studies, hypertension was the most common condition in all instances.

## 3. COVID-19 as a Risk Factor of Atherosclerosis or CVD

COVID-19 itself can also induce cardiovascular manifestations including myocardial injury, myocarditis, arrhythmias, acute coronary syndrome, and thromboembolism [20]. On the one hand, Hessami et al. conducted a systematic review and meta-analysis that included studies with hospitalized symptomatic COVID-19 patients. The authors found a high burden of CVDs among COVID-19 patients in different countries and a high prevalence of specific cardiovascular complications in this population, such as hypertension, myocardial damage, acute cardiac injury, arrhythmia, coronary artery disease, heart failure, valve heart disease, cardiomyopathy, and heart palpitation. Nevertheless, the study was unable to determine the causal relationship between COVID-19 and cardiovascular complications because there could be pre-existing conditions in patients or either developed by the infection [33]. On the other hand, the study of Pillarisetti et al. analyzed data of the TriNetX COVID-19 global research network, a cohort of 81,844 patients with a diagnosis of COVID-19. The study found that 9.3% of patients developed cardiac complications after diagnosis of COVID-19. Heart failure, atrial fibrillation, sinus bradycardia, and acute coronary syndrome were the most incident. Death occurred in 20% of patients with cardiac complications. Although the analysis was adjusted for sex and greater number of comorbidities, the mortality was significantly higher in that group than in the one without cardiac complications [49].

Table 2 includes the incidence of several CVDs in individuals hospitalized with COVID-19 analyzed in two different systematic reviews and meta-analyses. The results were heterogeneous, with arrythmia being the most common complication in one of them [31] and heart failure in the other [8].

## 4. Long-Term Cardiovascular Outcomes of COVID-19

Long COVID is defined as the continuation or development of new symptoms three months after the initial SARS-CoV-2 infection, with these symptoms lasting for at least 2 months with no other explanation [50]. Currently, three years after the declaration of the pandemic, emerging long-term outcome data demonstrate a significant burden of CVD following acute infection with SARS-CoV-2 [51]. Maestre-Muñiz et al. followed 581 surviving adult patients for 1 year who were discharged after hospital admission due to acute COVID-19. New-onset hypertension and de novo heart failure were present in 2% of the patients, and there was an increased need for readmission [52]. Raisei-Estabragh et al. found that in a matched cohort with 17,871 COVID-19 cases followed until the first episode of a specific outcome (heart failure, atrial fibrillation, venous thromboembolism, pericarditis, and mortality), most events occurred in the early post-infection period, typically within the first 30 days. The increased risk of the outcomes remained statistically significant after 30 days but with a smaller effect size [24]. Negreira-Caamaño et al. followed up on 673 patients for 1 year after discharge for COVID-19 hospitalization. Most major cardiovascular events (acute coronary syndrome, cerebrovascular event, venous thromboembolic disease, hospitalization for heart failure, and cardiovascular death) occurred more than one month after hospitalization, although 75% of the venous thromboembolic disease cases occurred within the first 30 days [53]. Xie et al. studied a cohort of 153,760 veterans who survived COVID-19 comparing it with two different cohorts, one of 5,859,411 noninfected patients during 2017 and another of 5,637,647 patients with no evidence of SARS-CoV-2. The incident rates of CVDs during the 12-month follow-up after infection were significantly higher than those in the pre-exposure period, showing that post-COVID patients had an increased risk of a broad range of CVDs (e.g., cerebrovascular disorders, dysrhythmias, ischemic and nonischemic heart disease, pericarditis, myocarditis, heart failure, and thromboembolic disease) [25]. Wang et al. followed 4,131,717 subjects from 30 days to 12 months after a test of SARS-CoV-2 had been performed. Compared to non-COVID-19 controls (n = 2,249,533), the 12-month risk of incidental CVDs (including cerebrovascular diseases, inflammatory heart disease, ischemic heart disease, and other cardiac disorders, such as heart failure and thromboembolic disorders) was substantially higher in the COVID-19 survivors (n = 691,455) [54]. Knight et al. took a population of 44,964,486 people and compared COVID-19-diagnosed patients hospitalized (118,879) and nonhospitalized (1,248,180) vs. non-COVID-19 patients. The authors found that 1 to 2 weeks after diagnosis there was a substantial increase in the relative incidence of arterial thrombotic and venous thromboembolic events. Although there was a decline with time since diagnosis, the incidence of venous thromboembolic events persisted for up to 49 weeks after diagnosis [55]. On the other hand, Rezel-Potts et al. followed a matched cohort of 428,650 COVID-19 patients without diabetes mellitus and CVD for 1 year after infection date [56]. The incidence of CVD increased in the first 3 months after COVID-19 diagnosis, while the incidence of diabetes mellitus remained elevated for at least 12 weeks [56]. All this evidence might point to the fact that post-COVID-19 sequelae, such as worsening hypertension, diabetes, renal damage, inflammation, induced dyslipidemia, and endothelial dysfunction, could contribute to early atherosclerosis, accelerate vascular aging, and arterial stiffness [17] (Figure 1).

## 5. How the Most Common Treatments for CVD Prevention Influence SARS-CoV-2 Infection

Because CVD is a common comorbidity in COVID-19 patients, CVD pharmacological treatments for primary and secondary prevention are commonly used in this population. Thus, from the beginning of the COVID-19 pandemic, CVD treatments were pointed to influence COVID-19 outcomes [57,58]. The fact that patients with CVD have a higher expression of ACE2 led to the suspicion that angiotensin-converting enzyme inhibitors (ACEIs) and angiotensin receptor blockers (ARBs) had an unfavorable influence. Another example of CVD treatment influence is that statins for the pleiotropic effect that lead to a reduction in inflammatory response, were hypothesized to be useful in COVID-19 [21]. Finally, coagulopathy is also common in persons infected with SARS-CoV-2, being a prevalent feature of atherosclerosis [22]. Thus, anticoagulants and aspirin might be potentially useful drugs for COVID-19 [59]. Moreover, pharmacological treatments with deleterious cardiovascular effects have not worked for COVID-19. Chloroquine/hydroxychloroquine and azithromycin have been associated with prolonged QT; azithromycin and remdesivir with conduction defects, ventricular arrhythmias, and heart failure; and lopinavir/ritonavir and interleukin therapies with ischemic heart disease and abnormalities of the lipid profile [9]. The interactions between nirmatrelvir-ritonavir (NMVr) and cardiovascular drugs in patients with COVID-19 have been recently described. Ritonavir, the pharmaceutical enhancer used in NMVr, is an inhibitor of the enzymes of the CYP450 pathway, particularly CYP3A4 and to a lesser degree CYP2D6, and affects the P-glycoprotein pump. Coadministration of NMVr with medications commonly used to manage cardiovascular conditions can potentially cause significant drug–drug interactions and may lead to severe adverse effects [60].

### 5.1. ACEIs and ARBs

Some initial evidence showed that a higher percentage of individuals with COVID-19 received either ACEIs or ARBs than those with severe or critical COVID-19. Particularly, in the study conducted by Feng et al., 87.9% of individuals with moderate COVID-19 were treated with ACEIs or ARBs, whereas this percentage was 6.1% for those in the severe and critical COVID-19 categories [26]. This observation raised the suspicion of a protective effect of such drugs. Indeed, given the relationship between the ACE2 receptor and SARS-CoV-2, ACEIs and ARBs could have an important role in the disease. Thus, the use of such drugs could potentially increase viral entry into cells. Both treatments are widely used for hypertension, and discontinuing or adding these treatments has been a difficult question to answer during the pandemic because of the lack of evidence available relating to the benefit or harm that could result [20,28,61]. The results regarding these active principles have been controversial, with some of them showing a decreased risk of severe disease or death by COVID-19 in those treated with ACEIs or ARBs [62,63], whereas, other studies including randomized controlled trials, showed no significant improvement or deleterious effect in SARS-CoV-2 infection [32,64,65,66,67,68].

ACEIs are excreted unchanged in the urine, translating into no significant interactions with NMVr, making them safe to continue [60]. However, ritonavir could reduce the antihypertensive effects of irbesartan, and, in contrast, the coadministration of NMVr with losartan can lead to hypotension [69]. The weak inhibition of the hepatic uptake transporter by NMVr may increase the concentration of both valsartan and the active metabolite of sacubitril, warranting close blood pressure monitoring [70].

Table 3 shows the most recent meta-analysis results on the effects of ACEIs and ARBs published during the last year. As previously summarized, the results were heterogeneous with half of the meta-analyses included showing a protective effect of such drugs on mortality and disease severity.

### 5.2. Statins

Statins have been thought to confer a potential advantage against progression to severe COVID-19 infection with different proposed effects including anti-inflammatory, antithrombotic, antiviral protease, and protection against endothelial dysfunction [75]. Thus, at the beginning of the pandemic, the evidence pointed out that statin use was associated with improved survival [76] and seemed to lower the incidence of severe COVID-19 and mortality rates [22]. So far, several publications showed that statin use could even decrease the risk of severe COVID-19 outcomes [77,78,79,80,81,82,83,84,85]. Overall, statin usage in Western patients hospitalized with COVID-19 was associated with nearly 40% lower odds of progressing toward severe illness or death. These estimates were observed after excluding studies in which statin therapy was started during hospital admission [80]. In addition, prior statin treatment was associated with a reduced risk of COVID-19 test positivity, diagnosis, hospitalization, and mortality in three different cohorts that included 572,695 patients followed up until Jan 2021 in Sweden [85]. Finally, in a cohort of 146,413 hospitalized COVID-19 patients, those who had a continuous (home and hospital) use of atorvastatin had a 35% reduction in the odds of mortality compared to patients who received only atorvastatin at home [86]. Nevertheless, the results from randomized controlled trials were not as conclusive as those observed in the cohort studies [87,88]. Thus, in the RESIST trial, the treatment with atorvastatin did not prevent clinical deterioration [87]. Additionally, the use of atorvastatin increased the length of stay at the hospital and frequency of ICU admission in patients with COVID-19 [88], and Matli et al. observed no improvement in the clinical outcomes in those with COVID-19 treated with atorvastatin [89]. Finally, and according to Hejazi et al., atorvastatin significantly reduced the supplemental oxygen need, hospitalization duration, and serum high-sensitivity C reactive protein (hs-CRP) in mild to moderate hospitalized COVID-19 patients [90]. The coadministration of ritonavir/saquinavir and simvastatin increased its 24-h area under the curve (AUC) by 3059% [91]. Therefore, NMVr should not be administered with simvastatin or lovastatin owing to the risk of myopathy and rhabdomyolysis [60].

Table 4 includes the most recent meta-analysis on the effect of statins published during the last year. The results confirm the protective effect of statins on COVID-19 mortality and severity.

### 5.3. Anticoagulants and Aspirin

At the beginning of the pandemics and due to the concerns of an extended prothrombotic risk with COVID-19, the use of extended thromboprophylaxis was considered although there were no clinical trial data [93]. On the one hand, Nadkarni et al. found a lower risk of mortality and intubation in individuals treated with anticoagulant drugs [94]. In hospitalized COVID-19 patients, no association was found between the intensity of thromboprophylaxis and the rate of thrombotic events although the rate of clinically relevant bleeding complications among patients who received intermediate- or full-dose anticoagulation exceeded that recorded among those treated with preventive doses [95]. Finally, in young to middle-aged patients with few comorbidities and with a predominantly mild disease, prophylactic anticoagulation was not recommended due to no venous thromboembolism event during post-hospitalization [96].

Regarding the interaction of anticoagulants with COVID-19 drugs, particularly NMVr, when direct oral anticoagulants cannot be interrupted or dose-adjusted, NMVr should not be administered. Complete withdrawal of anticoagulation while on NMVr should be reserved for extremely low-risk patients, knowing that COVID-19 infection can itself increase the risk of a thromboembolic event [60].

Recent data suggest that aspirin use is associated with improved COVID-19 patient outcomes, including a decreased risk of thromboembolism, mechanical ventilation, ICU admission, and mortality [97,98,99,100,101,102,103]. Aspirin has been proposed to act on the intracellular signaling pathway involved with viral replication and reduction in systemic inflammation, cytokine release, platelet activity, and hypercoagulability [22,104] (Figure 1).

No significant interactions are expected between aspirin and NMV use. Aspirin is metabolized by hepatic conjugation with glycin or glucuronic acid. Ritonavir weakly induces glucuronidation and can theoretically increase aspirin metabolism. However, there are no reported clinical adverse events, and it is safe to take it with NMVr [60,105].

Table 5 includes the results of the most recent meta-analysis on the effect of anticoagulants and aspirin. The results confirm the protective effect of the latter on COVID-19 mortality, whereas no significant effect has been found for the first.

## 6. Summary and Future Research

This review provides the latest insights into the interaction between COVID-19 and atherosclerosis and the cardiovascular event caused by these vascular injuries. Indeed, both atherosclerosis and COVID-19 present a bidirectional association. A history of CVD is considered a major risk factor for COVID-19 disease, and the follow-up of individuals with COVID-19 has shown that the disease increases the risk of CVD events. The pathophysiological effects of both diseases (e.g., inflammation, immune response, and endothelial damage) have been proposed as the main potential mechanisms behind this bidirectional interplay. Moreover, research works have identified the interaction between several CVD treatments that might play a role in preventing COVID-19 complications.

The primary and secondary prevention of CVD is crucial in clinical practice for three reasons. First, CVD is the leading cause of mortality in the world and continues to increase in low- and lower-middle-income countries. Second, noncommunicable diseases, such as CVDs, are characterized by a long induction period that is generally asymptomatic. Indeed, its first manifestation is frequently a vital event, such as an acute myocardial infarction or a stroke. Finally, the control of risk factors, that is, factors associated with CVD, leads to a reduction in its incidence. The control of CVD, whose morbidity and mortality are very high, will have an impact not only on the individual at risk, but also on the population overall, as many individual attitudes are shaped by the community’s attitude toward health problems. Thus, an accurate and reliable identification of the individual risk is imperative to decrease the incidence of CVD. To improve the estimation of such risk, a better understanding of the link among atherosclerosis, CVD, and COVID-19 is vital. As a result, public health strategies will be developed to improve the prognosis for patients with CVD and COVID-19 or to mitigate the short-, mid- and long-term cardiovascular outcomes in patients with COVID-19. Therefore, more high-quality scientific works exploring the bidirectional link between atherosclerosis–CVD and COVID-19 are required to improve the knowledge in this field.

## Figures and Tables

**Figure 1 biomedicines-11-01206-f001:**
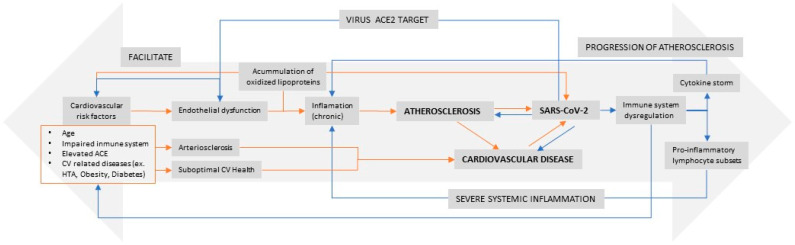
Bidirectional association between atherosclerosis and SARS-CoV2 infection (adapted from Vinciguerra et al. [27]).

**Table 1 biomedicines-11-01206-t001:** Prevalence and range (%) of cardiovascular risk factors and CVDs in hospitalized patients with COVID-19.

	Pellicori [31]	Yang [47]	Emami [48]
Cardiovascular risk factors			
Hypertension	36.1% (4.5% to 100%)	21.1% (13.0% to 27.2%)	16.4% (10.2% to 23.7%)
Obesity	21.6% (0.2% to 57.6%)	--	--
Diabetes	22.1% (0.0% to 100%)	9.7% (7.2% to 12.2%)	7.9% (6.6% to 9.3%)
CVDs			
Ischemic heart disease	22.1% (0.0% to 100%)	--	--
Cardiovascular disease	23.5% (0.7% to 68.7%)	8.4% (3.8% to 13.8%)	12.1% (4.4% to 22.8%)
Heart failure	6.5% (0.0% to 28.0%)	--	--
Cerebrovascular accident	5.1% (0.5% to 19.6%)	--	--
Atrial fibrillation	11.1% (1.0% to 22.8%)	--	--
Valve disease	3.7% (1.8% to 6.8%)	--	--

**Table 2 biomedicines-11-01206-t002:** Incidence of cardiovascular events in patients with COVID-19.

	Pellicori [31]	Kunutsor [8]
Myocardial infarction/Acute Coronary syndrome	1.7% (0.0% to 3.6%)	6.2% (1.8% to 12.3%)
Stroke	1.2% (0.0% to 9.6%)	1.6% (0.6% to 4.7%)
Heart failure	6.8% (0.0% to 24.0%)	17.6% (14.2% to 21.2%)
Venous thromboembolism	7.4% (0.0% to 46.2%)	1.6% (0.6% to 4.7%)
Coagulopathy	8.0% (0.5% to 38.0%)	--
Arrythmia	9.3% (0.0% to 30.3%)	9.3% (5.1% to 14.6%)

**Table 3 biomedicines-11-01206-t003:** Odds ratio (95% confidence interval) of the effects of ACEIs and ARBs on COVID-19 outcomes.

	Mortality	Severity *
Gnanenthiran [71]	0.95 (0.69 to 1.30)	1.00 (0.77 to 1.30)
Huang [72]	0.61 (0.52 to 0.70)	0.99 (0.83 to 1.17)
Kurdi [73]	0.80 (0.75 to 0.86)	0.86 (0.78 to 0.95)
Lee [74]	0.75 (0.61 to 0.92)	0.80 (0.58 to 1.10)
Liu [63]	0.80 (0.41 to 1.57)	0.62 (0.44 to 0.88)

* Hospital or ICU admission.

**Table 4 biomedicines-11-01206-t004:** Odds ratio (95% confidence interval) of the effects of statins on COVID-19 outcomes.

	Mortality	Severity *
Lao [78]	0.72 (0.67 to 0.77)	0.94 (0.89 to 0.99)
Pal [79]	0.51 (0.41 to 0.63)	1.02 (0.69 to 1.50)
Zein [92]	0.72 (0.55 to 0.95)	--

* Hospital or ICU admission.

**Table 5 biomedicines-11-01206-t005:** Odds ratio (95% confidence interval) of the effects of anticoagulants and aspirin on COVID-19 outcomes.

Anticoagulants	Mortality	Severity *
Reis [106]	1.03 (0.86 to 1.24)	--
Zeng [107]	1.08 (0.90 to 1.30)	1.50 (0.72 to 3.12)
Aspirin		
Kow [108]	0.50 (0.32 to 0.77)	--
Martha [101]	0.46 (0.35 to 0.61)	--
Salah [109]	1.12 (0.84 to 1.50)	--
Srivastava [110]	0.70 (0.63 to 0.77)	--

* Hospital or ICU admission.

## Data Availability

Not applicable.

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
