# Peer review of "Atherosclerosis, Cardiovascular Disease, and COVID-19: A Narrative Review"

_biomedicines, 2023, doi:10.3390/biomedicines11041206_

Round 1

Reviewer 1 Report

Atherosclerosis is a chronic inflammatory and degenerative process that mainly occurs in arteries and is morphologically characterized by asymmetric focal thickenings of the intima of the artery. This process is the basis of cardiovascular diseases (CVD), the most common cause of death worldwide. Some studies suggest a bidirectional link between atherosclerosis and the consequent CVD, with COVID-19. Evidence shows that the COVID-19 prognosis in individuals with CVD is worse compared with those without. Various studies have reported the emergence of newly diagnosed patients with CVD after COVID-19. In this review article, the authors briefly discussed their implication in the infection process; a better understanding of the link between atherosclerosis, CVD and COVID-19 could proactively identify risk factors and, as a result, develop strategies to improve the prognosis for these patients. It is an interesting review article. Specific comments follow:

1.          The iconographies and tables are particularly welcome for the review article to attract readers. I suggest the author include iconographies [such as the pathophysiological effects of both diseases (e.g. inflammation, immune response and endothelial damage) that have been proposed as the main potential mechanisms behind this bidirectional interplay] to attract readers to the review article. Because icons and tables can make the article more visually appealing and help readers better understand the content. Additionally, they can help readers quickly skim through the article to understand the main points and conclusions.

2.          Please update the accessed date of Reference #1.

3.          Are there any potential side effects of cardiovascular drugs on COVID-19 outcomes?

4.          The authors should discuss the severe limitations of current studies, and give future directions for research in the field of atherosclerosis, CVD and COVID-19.

Minor:

1. Line 50: Itis  ->  It is

2. Line 272: [86,87,88,89,90,91,92]  ->  [86-92]

3. Line 286: disease and  the follow-up  ->  disease and the follow-up

Author Response

Reviewer 1
Atherosclerosis is a chronic inflammatory and degenerative process that mainly occurs in arteries and is morphologically characterized by asymmetric focal thickenings of the intima of the artery. This process is the basis of cardiovascular diseases (CVD), the most common cause of death worldwide. Some studies suggest a bidirectional link between atherosclerosis and the consequent CVD, with COVID-19. Evidence shows that the COVID-19 prognosis in individuals with CVD is worse compared with those without. Various studies have reported the emergence of newly diagnosed patients with CVD after COVID-19. In this review article, the authors briefly discussed their implication in the infection process; a better understanding of the link between atherosclerosis, CVD and COVID-19 could proactively identify risk factors and, as a result, develop strategies to improve the prognosis for these patients. It is an interesting review article. Specific comments follow:

  1. The iconographies and tables are particularly welcome for the review article to attract readers. I suggest the author include iconographies [such as the pathophysiological effects of both diseases (e.g. inflammation, immune response and endothelial damage) that have been proposed as the main potential mechanisms behind this bidirectional interplay] to attract readers to the review article. Because icons and tables can make the article more visually appealing and help readers better understand the content. Additionally, they can help readers quickly skim through the article to understand the main points and conclusions.

Reply: We have included a new figure 1, where we explain the bidirectional association between atherosclerosis (and the consequent cardiovascular disease) and SARS-CoV-2 infection.

Figure 1. Bidirectional association between atherosclerosis and SARS-CoV2 infection (adapted from Vinciguerra et al. (Vinciguerra M. J Clin Med. 2020).

We have added a new reference:

Vinciguerra, M.; Romiti, S.; Fattouch, K.; De Bellis, A.; Greco, E. Atherosclerosis as Pathogenetic Substrate for Sars-Cov2 Cytokine Storm. J. Clin. Med. 2020;9:2095. doi: 10.3390/jcm9072095.

  1. Please update the accessed date of Reference #1.

Reply: We have updated the accessed date of Reference #1.

  1. Are there any potential side effects of cardiovascular drugs on COVID-19 outcomes?

Reply: The pharmacological treatments with deleterious cardiovascular effects have not worked for COVID-19. Examples like chloroquine/hydroxychloroquine and azithro-mycin have been associated with prolonged QT, azithromycin and remdesivir with conduction defects, ventricular arrhythmias and heart failure and lopinavir/ritonavir and interleukin therapies with ischemic heart disease and abnormalities of the lipid profile (Aggarwal, G. Curr. Probl. Cardiol. 2020). Nevertheless, a recent manuscript has described interactions between nirmatrelvir-ritonavir (NMVr) and cardiovascular drugs in patients with COVID-19. Ritonavir, the pharmaceutical enhancer used in NMVr, is an inhibitor of the enzymes of CYP450 pathway, particularly CYP3A4 and to a lesser degree CYP2D6, and affects the P-glycoprotein pump. Co-administration of NMVr with medications commonly used to manage cardiovascular conditions can potentially cause significant drug-drug interactions and may lead to severe adverse effects (Abraham, S. J Am Coll Cardiol. 2022). We have discussed this point in paragraph 5 and in subparagraphs 5.1, 5.2 and 5.3 of the new version of the manuscript.

Page 6, 1st paragraph. (5. How CVD treatments influence SARS-CoV-2 infection)

“The interactions between nirmatrelvir-ritonavir (NMVr) and cardiovascular drugs in patients with COVID-19 has been recently described. Ritonavir, the pharmaceutical enhancer used in NMVr, is an inhibitor of the enzymes of CYP450 pathway, particularly CYP3A4 and to a lesser degree CYP2D6, and affects the P-glycoprotein pump. Co-administration of NMVr with medications commonly used to manage cardiovascular conditions can potentially cause significant drug-drug interactions and may lead to severe adverse effects (Abraham, S. J Am Coll Cardiol. 2022)”.

Page 6, 3rd paragraph. (5.1 ACEI & ARBs)

“ACEI are excreted unchanged in the urine, translating into no significant interactions with NMVr, making them safe to continue (Abraham, S. J Am Coll Cardiol. 2022). However, ritonavir could reduce the antihypertensive effects of irbesartan and, in contrast, the co-administration of NMVr with losartan can lead to hypotension (Fichtenbaum CJ. Clin Pharmacokinet. 2002). Weak inhibition of the hepatic up-take transporter by NMVr may increase the concentration of both valsartan and the active metabolite of sacubitril, warranting close blood pressure monitoring (Hanna I. Xenobiotica. 2018)”.

Page 7, 2nd paragraph. (5.2 Statins)

“The co-administration of ritonavir/saquinavir and simvastatin increases its 24-hour area under the curve (AUC) by 3,059% (Wiggins BS. Am J Cardiovasc Drugs. 2017). Therefore, NMVr should not be administered with simvastatin or lovastatin owing to risk of myopathy and rhabdomyolysis (Abraham, S. J Am Coll Cardiol. 2022)”.

Page 8, 2nd paragraph. (5.3 Anticoagulants & aspirin)

“Regarding the interaction of anticoagulants with COVID-19 drugs, particularly NMVr, when direct oral anticoagulants cannot be interrupted or dose adjusted, NMVr should not be administered. Complete withdrawal of anticoagulation while on NMVr should be reserved for extremely low-risk patients, knowing that COVID-19 infection can itself increase the risk of a thromboembolic event (Abraham, S. J Am Coll Cardiol. 2022)”.

Page 8, 4th paragraph 

“No significant interactions are expected between aspirin and NMVs. Aspirin is metabolized by hepatic conjugation with glycin or glucuronic acid. Ritonavir weakly induces glucuronidation and can theoretically increase aspirin metabolism. However, there are no reported clinical adverse events, and it is safe to take with NMVr (Abraham, S. J Am Coll Cardiol. 2022; Liverpool Drug Interactions Group (https://www.covid19-druginteractions.org/)”.

We have included five new references:

Abraham, S.; Nohria, A; Neilan, T.G.; Asnani, A.; Saji, A.M.; Shah, J.; Lech, T.; Grossman, J.; Abraham, G.M.; McQuillen, D.P.; Martin, D.T.; Sax, P.E.; Dani, S.S.; Ganatra, S. Cardiovascular drug interactions with nirmatrelvir/ritonavir in patients with COVID-19: JACC review topic of the week. J. Am. Coll. Cardiol. 2022, 80; 1912-1924. doi: 10.1016/j.jacc.2022.08.800.

Fichtenbaum, C.J., Gerber, J.G. Interactions between antiretroviral drugs and drugs used for the therapy of the metabolic complications encountered during HIV infection. Clin. Pharmacokinet. 2002, 41, 1195-211. doi: 10.2165/00003088-200241140-00004.

Hanna, I., Alexander, N., Crouthamel, M.H., Davis, J., Natrillo, A., Tran, P., Vapurcuyan, A., Zhu, B. Transport properties of valsartan, sacubitril and its active metabolite (LBQ657) as determinants of disposition. Xenobiotica 2018, 48, 300-313. doi: 10.1080/00498254.2017.1295171.

Wiggins, B.S., Lamprecht, D.G. Jr., Page, R.L. 2nd, Saseen, J.J. Recommendations for Managing Drug-Drug Interactions with Statins and HIV Medications. Am. J. Cardiovasc. Drugs 2017, 17, 375-389. doi: 10.1007/s40256-017-0222-7.

Liverpool Drug Interactions Group. Interactions with essential medicines and nirmatrelvir/ritonavir. Available online: https://www.covid19-druginteractions.org/ (Accessed March 12, 2023).

  1. The authors should discuss the severe limitations of current studies, and give future directions for research in the field of atherosclerosis, CVD and COVID-19.

Reply: We have included a paragraph about the limitations of the current studies and future directions of research in Section 6, now entitled “Summary and future research” (page 8, 7th paragraph): “The primary prevention of CVD is crucial in clinical practice for 3 reasons. First, CVD is the leading cause of mortality in the world, and continues to increase in low and lower middle-income countries. Second, the long induction period—generally asymptomatic—of atherosclerosis means that its first manifestation is frequently a vital event such as an acute myocardial infarction or stroke. Finally, the control of risk factors, that is, factors associated with this disease, leads to a reduction in its incidence. The control of CVD, whose morbidity and mortality are very high, will have an impact not only on the individual at risk, but on the population as a whole, as many individual attitudes are shaped by the community’s attitude toward health problems. Thus, an accurate and reliable identification of the individual risk is imperative to decrease the incidence of CVD. A better understanding of the link between atherosclerosis, CVD and COVID-19 could proactively identify risk factors and, as a result, develop strategies to improve the prognosis for these patients. Therefore, more high-quality scientific works exploring the bidirectional link between atherosclerosis – CVD and COVID-19 are required to improve the knowledge on this field”. 

Minor:

  1. Line 50: Itis  ->  It is

  1. Line 272: [86,87,88,89,90,91,92]  ->  [86-92]

  1. Line 286: disease and  the follow-up  ->  disease and the follow-up

Reply: We have corrected all typos

Reviewer 2 Report

The scientific value of the article is low.

The subject, although an interesting one, is treated very superficially without getting into the essence of the physiopathological mechanisms of the atherosclerotic phenomenon with the particularities in the Covid19 disease. More details about Atherosclerosis as a morphopathological phenomen are needed.

The difference between atherosclerosis as a risk factor and the clinical manifestations of atherosclerosis as comorbidities associated with the Covid 19 disease is not clear.

Several classes of drugs are listed, but not all of those with an impact on the evolution of the atheroma plaque.

It would have been interesting to mention  the diagnostic methods of subclinical atherosclerosis (those that evaluate the atheroma plaque).

Thus, in this form of the article, the novelty elements are missing.

Some of my comments are inserted into the text

Author Response

Reviewer 2

The scientific value of the article is low.

The subject, although an interesting one, is treated very superficially without getting into the essence of the physiopathological mechanisms of the atherosclerotic phenomenon with the particularities in the Covid19 disease. More details about Atherosclerosis as a morphopathological phenomen are needed.

Reply: The new version of the manuscript includes a Figure that shows the bidirectional link between atherosclerosis and COVID-19.

Figure 1. Bidirectional association between atherosclerosis and SARS-CoV2 infection (adapted from Vinciguerra et al. (Vinciguerra M. J Clin Med. 2020).

We have added a new reference:

Vinciguerra, M.; Romiti, S.; Fattouch, K.; De Bellis, A.; Greco, E. Atherosclerosis as Pathogenetic Substrate for Sars-Cov2 Cytokine Storm. J. Clin. Med. 2020;9:2095. doi: 10.3390/jcm9072095.

We have also added a new paragraph in Introduction (page 2, 1st paragraph): “A pro-inflammatory and thrombophilic state is an integral feature of atherosclerosis, potentially increasing vulnerability to severe COVID-19 because the underlying endo-thelial dysfunction might represent the ideal deregulated immunological setting in which SARS-CoV-2 triggers a “cytokine storm” (Makarova YA. Diagnostics (Basel). 2023)”.

We have added a new reference:

Makarova, Y.A.; Ryabkova, V.A.; Salukhov, V.V.; Sagun, B.V.; Korovin, A.E.; Churilov, L.P. Atherosclerosis, Cardiovascular Disorders and COVID-19: Comorbid Pathogenesis. Diagnostics (Basel). 2023, 13, 478. doi: 10.3390/diagnostics13030478.

The difference between atherosclerosis as a risk factor and the clinical manifestations of atherosclerosis as comorbidities associated with the Covid 19 disease is not clear.

Reply: In Table 1, we have presented the results of the meta-analysis that estimate the incidence of several cardiovascular risk factors in individuals hospitalized with COVID-19. We agree with the reviewer that some of the variables presented as CVD risk factors are established CVD (e.g. heart failure, ischemic heart disease…). Therefore, we have modified the title and structure of Table 1 for the sake of accuracy.

Table 1. Prevalence and range (%) of cardiovascular risk factors and CVDs in hospitalized patients with COVID-19

Pellicori

Yang

Emami

Cardiovascular risk factors

Hypertension

36.1%

(4.5% to 100%)

21.1%

(13.0% to 27.2%)

16.4%

(10.2% to 23.7%)

Obesity

21.6%

(0.2% to 57.6%)

--

--

Diabetes

22.1%

(0.0% to 100%)

9.7%

(7.2% to 12.2%)

7.9%

(6.6% to 9.3%)

CVDs

Ischemic heart disease

22.1%

(0.0% to 100%)

--

--

Cardiovascular disease

23.5%

(0.7% to 68.7%)

8.4%

(3.8% to 13.8%)

12.1%

(4.4% to 22.8%)

Heart failure

6.5%

(0.0% to 28.0%)

--

--

Cerebrovascular accident

5.1%

(0.5% to 19.6%)

--

--

Atrial fibrillation

11.1%

(1.0% to 22.8%)

--

--

Valve disease

3.7%

(1.8% to 6.8%)

--

--

Several classes of drugs are listed, but not all of those with an impact on the evolution of the atheroma plaque.

Reply: We have added the most common used drugs for the secondary prevention of CVD (ACEI, statins and aspirin). Indeed, this combination has been promoted by several doctors in a simple pill (i.e. the polypill) in order to increase the adherence to such treatments (Fuster V. Nat Clin Pract Cardiovasc Med. 2007; Castellano JM. J Am Coll Cardiol. 2014; Tamargo J. Int J Cardiol. 2015). In addition, the body of evidence on the effect of such drugs on COVID-19, is larger than that published for other active principles that also have a role on the evolution of the atherosclerotic plaque.

It would have been interesting to mention the diagnostic methods of subclinical atherosclerosis (those that evaluate the atheroma plaque).

Reply: We have included a paragraph discussing subclinical atherosclerosis and COVID-19 in the new version of the manuscript (page 3, 3rd paragraph): “Additionally, subclinical atherosclerosis can impact the course of COVID-19. Coronary artery calcification, a specific imaging marker of coronary atherosclerosis that correlates with the plaque burden, can reveal previously undiagnosed CVD in COVID-19 patients (Budoff MJ. Circulation. 2017). In a retrospective study of 457 individuals without history of clinical coronary artery disease who underwent chest CT imaging during COVID-19 hospitalization, coronary artery calcification was detected in 42.9%. Presence of coronary artery calcification was associated with mechanical ventilation (p = 0.01), ICU admission (p = 0.02), in-hospital or 30-day mortality (p < 0.01), and hospital length of stay (p < 0.001) (Kotlo S. Am Heart J Plus. 2023). Moreover, a recent meta-analysis that included 8 studies, showed an increase in mortality in the presence of coronary artery calcifications (relative risk 2.24, 95% confidence interval, 1.41–3.56; p < 0.001) (Lee KK. Med Sci (Basel). 2022)”.

We have added three new references:

Budoff, M.J.; Mayrhofer, T.; Ferencik, M.; Bittner, D.; Lee, K.L.; Lu, M.T.; Coles, A.; Jang, J.; Krishnam, M.; Douglas, P.S.; Hoffmann, U.; PROMISE Investigators. Prognostic Value of Coronary Artery Calcium in the PROMISE Study (Prospective Multicenter Imaging Study for Evaluation of Chest Pain). Circulation 2017, 136, 1993-2005. doi: 10.1161/CIRCULATIONAHA.117.030578.

Kotlo, S.; Thorgerson, A.; Kulinski, J. Coronary artery calcification as a predictor of adverse outcomes in patients hospitalized with COVID-19. Am. Heart J. Plus 2023, 28, 100288. doi: 10.1016/j.ahjo.2023.100288.

Lee, K.K.; Rahimi, O.; Lee, C.K.; Shafi, A.; Hawwass, D. A Meta-Analysis: Coronary Artery Calcium Score and COVID-19 Prognosis. Med. Sci. (Basel). 2022, 10, 5. doi: 10.3390/medsci10010005.

Thus, in this form of the article, the novelty elements are missing.

Some of my comments are inserted into the text

Reply: We thank the reviewer for all comments that have helped to improve the quality of the manuscript.

Reviewer 3 Report

The manuscript biomedicines-2283782 entitled “Atherosclerosis, Cardiovascular Disease and COVID-19: A Narrative Review” by Carles Vilaplana-Carnerero  and coworkers is a review about atherosclerosis and COVID19. Some studies suggest a bidirectional link between atherosclerosis and the consequent CVD, with COVID-19. Indeed, a growing body of evidence shows that COVID-19 prognosis in individuals with CVD is worse compared with those without. Moreover, various studies have reported the emergence of newly diagnosed patients with CVD after COVID-19. The most common treatments for CVD may influence COVID-19.

The review work was well planned and well written.

The revision of literature is complete.

It is missing a part about the changes in lipid profile related to COVID-19

Tables are clear.

Minor comment:

Line 50 itis should be corrected as it is

A linguistic revision is recommended.

Author Response

Reviewer 3

The manuscript biomedicines-2283782 entitled “Atherosclerosis, Cardiovascular Disease and COVID-19: A Narrative Review” by Carles Vilaplana-Carnerero and coworkers is a review about atherosclerosis and COVID19. Some studies suggest a bidirectional link between atherosclerosis and the consequent CVD, with COVID-19. Indeed, a growing body of evidence shows that COVID-19 prognosis in individuals with CVD is worse compared with those without. Moreover, various studies have reported the emergence of newly diagnosed patients with CVD after COVID-19. The most common treatments for CVD may influence COVID-19.

The review work was well planned and well written.

The revision of literature is complete.

It is missing a part about the changes in lipid profile related to COVID-19

Reply: We have included a brief paragraph about the changes in lipid profile related to COVID-19 (page 3, 1st paragraph): “Regarding the lipid profile, results from a meta-analysis showed significantly decreased levels of total cholesterol, high-density lipoprotein (HDL) cholesterol, and low-density lipoprotein (LDL) cholesterol are associated with severity and mortality in COVID-19 patients. Hence, the lipid profile may be used for assessing the severity and prognosis of COVID-19 (Mahat RK. Clin Nutr ESPEN. 2021)”.

We have added a new reference:

Mahat, R.K.; Rathore, V.; Singh, N.; Singh, N.; Singh, S-K.; Shah, R.K.; Garg, C. Lipid profile as an indicator of COVID-19 severity: A systematic review and meta-analysis. Clin. Nutr. ESPEN 2021, 45, 91-101. doi: 10.1016/j.clnesp.2021.07.023.

Tables are clear.

Reply: Thank you for the comment

Minor comment:

Line 50 itis should be corrected as it is

Reply: We have corrected the typo

A linguistic revision is recommended.

Reply: We have done a linguistic revision.

Reviewer 4 Report

The study, conducted by Carles Vilaplana-Carnerero et al., aimed to review recent studies that show a bidirectional relationship between COVID-19 and atherosclerosis, as well as to summarize the impact of cardiovascular drugs on COVID-19 outcomes.

Overall, this article is nicely written and has scientific value. However, some remarks should be clarified.

1. The Abstract of the manuscript should contain the aim of the work.

2. Please explain the data search process for this review.

3. Lines 184-186 - the sentence needs citation.

4. Lines 193-198 - citation needed.

5. Lines 209-210 - Sentence "Some preliminary evidence showed that a higher percentage of people with COVID-19 received an ACEI or ARB than those with severe or critical COVID-19." needs clarification and adding citation. A greater proportion of which COVID-19 patients received an ACEI or ARB compared to those with severe or critical COVID-19?

6. Table 4 - what about the below meta-analyses?

- Kow CS, Hasan SS. Meta-analysis of the effects of statins on COVID-19 patients. Am J Cardiol. November 1, 2020;134:153-155. doi: 10.1016/j.amjcard.2020.08.004. Or

- Lao US, Law CF, Baptista-Hon DT, Tomlinson B. Systematic review and meta-analysis of statin use and mortality, intensive care unit admissions, and mechanical ventilation requirements in COVID-19 patients. J Clin Med. Sep 16, 2022;11(18):5454. doi: 10.3390/jcm11185454.

Author Response

Reviewer 4

The study, conducted by Carles Vilaplana-Carnerero et al., aimed to review recent studies that show a bidirectional relationship between COVID-19 and atherosclerosis, as well as to summarize the impact of cardiovascular drugs on COVID-19 outcomes.

Overall, this article is nicely written and has scientific value. However, some remarks should be clarified.

  1. The Abstract of the manuscript should contain the aim of the work.

Reply: The objective of the manuscript has been added to the Abstract in the new version of the manuscript.

Abstract: Atherosclerosis is a chronic inflammatory and degenerative process that mainly occurs in large and medium-sized arteries and is morphologically characterized by asymmetric focal thickenings of the innermost layer of the artery, the intima. This process is the basis of cardiovascular diseases (CVD), the most common cause of death worldwide. Some studies suggest a bidirectional link between atherosclerosis and the consequent CVD, with COVID-19. The aims of this narrative review are: (1) to provide an overview of the most recent studies that point out a bidirectional relation between COVID-19 and atherosclerosis and, (2) to summarize the impact of cardiovascular drugs on COVID-19 outcomes. A growing body of evidence shows that COVID-19 prognosis in individuals with CVD is worse compared with those without. Moreover, various studies have reported the emergence of newly diagnosed patients with CVD after COVID-19. The most common treatments for CVD may influence COVID-19. Thus, their implication in the infection process is briefly discussed in this review. A better understanding of the link between atherosclerosis, CVD and COVID-19 could proactively identify risk factors and, as a result, develop strategies to improve the prognosis for these patients.

  1. Please explain the data search process for this review.

Reply: This is a narrative review, then, we did not perform a structured data search process. Nevertheless, we have reviewed the most updated data on the bidirectional association between atherosclerosis-CVD and COVID-19.

  1. Lines 184-186 - the sentence needs citation.

Reply: We have added the reference to the sentence: Rezel-Potts, E.; Douiri, A.; Sun, X;, Chowienczyk, P.J.; Shah, A.M.; Gulliford, M.C. Cardiometabolic outcomes up to 12 months after COVID-19 infection. A matched cohort study in the UK. PLoS. Med. 2022, 19, e1004052. doi: 10.1371/journal.pmed.1004052.

  1. Lines 193-198 - citation needed.

Reply: We have added two new references to the sentence: Task Force for the management of COVID-19 of the European Society of Cardiology. European Society of Cardiology guidance for the diagnosis and management of cardiovascular disease during the COVID-19 pandemic: part 1-epidemiology, pathophysiology, and diagnosis. Eur. Heart J. 2022, 43, 1033-1058. doi: 10.1093/eurheartj/ehab696.

Task Force for the management of COVID-19 of the European Society of Cardiology. ESC guidance for the diagnosis and management of cardiovascular disease during the COVID-19 pandemic: part 2-care pathways, treatment, and follow-up. Cardiovasc. Res. 2022, 118, 1618-1666. doi: 10.1093/cvr/cvab343.

  1. Lines 209-210 - Sentence "Some preliminary evidence showed that a higher percentage of people with COVID-19 received an ACEI or ARB than those with severe or critical COVID-19." needs clarification and adding citation. A greater proportion of which COVID-19 patients received an ACEI or ARB compared to those with severe or critical COVID-19?

Reply: We have added a brief explanation about this point (page 6, 2nd paragraph): “Some initial evidence showed that higher percentage of individuals with COVID-19 received either ACEI or ARB than those with severe or critical COVID-19. Particularly, in the study conducted by Feng et al. 87.9% of individuals with moderate COVID-19 were treated with ACEI or ARB; whereas this percentage was 6.1% for those in the with severe and critical COVID-19 categories (Feng Y. Am J Respir Crit Care Med. 2020).

In addition, we have added the reference to the sentence: Feng, Y.; Ling, Y.; Bai, T.; Xie, Y.; Huang, J.; Li, J.; Xiong, W.; Yang, D.; Chen, R.; Lu, F.; Lu, Y.; Liu, X.; Chen, Y.; Li, X.; Li, Y.; Summah, H.D.; Lin, H.; Yan, J.; Zhou, M.; Lu, H.; Qu, J. COVID-19 with different severities: a multicenter study of clinical features. Am. J. Respir. Crit. Care Med 2020, 201, 1380-1388. doi: 10.1164/rccm.202002-0445OC.

  1. Table 4 - what about the below meta-analyses?

- Kow CS, Hasan SS. Meta-analysis of the effects of statins on COVID-19 patients. Am J Cardiol. November 1, 2020;134:153-155. doi: 10.1016/j.amjcard.2020.08.004. Or

- Lao US, Law CF, Baptista-Hon DT, Tomlinson B. Systematic review and meta-analysis of statin use and mortality, intensive care unit admissions, and mechanical ventilation requirements in COVID-19 patients. J Clin Med. Sep 16, 2022;11(18):5454. doi: 10.3390/jcm11185454.

Reply: Thank you for the suggestion. Since we have included the most recent meta-analysis in Table 4, those conducted in 2022 or 2023, we have added Lao et al. study in Table 4 in the new version of the manuscript.

Round 2

Reviewer 2 Report

I have read the new version of the article. Some new ideas have been added but I don't feel that the topic has been fully debated.

I have inserted some of my comments direcly into the text.

The mechanisms of bidirectional association between atherosclerosis and SARS-CoV2 infection are more complex than are illustrated in figure nr 1.

The title of the section ”How CVD treatments influence SARS-CoV-2 infection” suggest that is about all the treatments used currently in cardiovascular diseases but only a few classes are discussed .

Several classes of drugs are listed, but not all of those with an impact on the evolution of the atheroma plaque.

Many of the ideas are repeated, without transmitting new scientific messages.

Author Response

Reviewer 2

I have read the new version of the article. Some new ideas have been added but I don't feel that the topic has been fully debated.

I have inserted some of my comments directly into the text.

Reply: Thank you for the comments:

Line 50: we have deleted the definition of cardiovascular diseases

Line 112: we have changed “dead” by “death”

Line 179: The section is now entitled: “4. Long-term cardiovascular outcomes of COVID-19”

Line 359: We have modified the conclusion to emphasize the bidirectional association between atherosclerosis, CVD and Sars-CoV-2 infection: “To improve the estimation of such risk, a better understanding of the link between atherosclerosis, CVD and COVID-19 is vital. As a result, public health strategies will be developed to improve the prognosis for patients with CVD and COVID-19, or to mitigate the short-, mid- and long-term cardiovascular outcomes in patients with COVID-19”.

The mechanisms of bidirectional association between atherosclerosis and SARS-CoV2 infection are more complex than are illustrated in figure in 1.

Reply: We agree with the reviewer. The Figure 1 presents the associations between atherosclerosis and SARS-CoV-2 infection schematically. Nevertheless, we have added a new version of Figure 1 in the new version of the manuscript. Moreover, the reference to Figure 1 has also been added in several parts of the manuscript where the association between both atherosclerosis and SARS-CoV-2 infection is explained in depth.

Figure 1. Bidirectional association between atherosclerosis and SARS-CoV2 infection (adapted from Vicinguerra et al. [Vinciguerra, M. J. Clin. Med. 2020])

The title of the section” How CVD treatments influence SARS-CoV-2 infection” suggest that is about all the treatments used currently in cardiovascular diseases but only a few classes are discussed.

Reply: We have modified the title of section 5: “5. How the most common treatments for CVD prevention influence SARS-CoV-2 infection”

Several classes of drugs are listed, but not all of those with an impact on the evolution of the atheroma plaque.

Reply: We agree with the reviewer. We have listed the most common drugs used in CVD prevention since we are seeking for the clinical utility of the manuscript. Indeed, the list includes the pharmacological components of the polypill that puts in a single pill aspirin, a statin and an angiotensin-converting enzyme inhibitor.   

Many of the ideas are repeated, without transmitting new scientific messages.

Reply: We have tried to write a well-balanced manuscript aimed to summarize the most recent knowledge of the bidirectional association between atherosclerosis – CVD and SARS-CoV-2 infection. Due to the broad audience of the journal, we have understood that several professionals will read the manuscript (basic researchers, clinicians and epidemiologists). Therefore, we have tried to adapt the message to this reality.

Round 3

Reviewer 2 Report

I have read the third version of the manuscript.

In my opinion the manuscript still has some inaccuracies related especially to the medical terminology used.

I have inserted some of my comments into the text. 

The manuscript needs to be improved in order to correspond with the aim of the research.

Author Response

Reviewer 2

I have read the third version of the manuscript.

In my opinion the manuscript still has some inaccuracies related especially to the medical terminology used.

I have inserted some of my comments into the text. 

The manuscript needs to be improved in order to correspond with the aim of the research.

Reply: Thank you for the comments. We have included the modifications proposed by the reviewer:

Line 100: we have modified the paragraph title, being “2. Atherosclerosis and CVD as a COVID-19 risk factor” in the new version of the manuscript

Line 222: we have modified the paragraph to explain that the treatments included can be used in either primary or secondary prevention: “Due to the fact that CVD is a common comorbidity in COVID-19 patients, CVD pharmacological treatments for primary and secondary prevention are commonly used in this population”.

Lines 359-360: the sentence has been reformulated: “Second, non-communicable diseases, such as CVD are characterized by a long induction period -generally asymptomatic-. Indeed, its first manifestation is frequently a vital event such as an acute myocardial infarction or a stroke”.

In addition, we have improved Figure 1 in the new version of the manuscript:
